# Highly Porous Expanded Graphite: Thermal Shock vs. Programmable Heating

**DOI:** 10.3390/ma14247687

**Published:** 2021-12-13

**Authors:** Alexander G. Bannov, Arina V. Ukhina, Evgenii A. Maksimovskii, Igor Yu. Prosanov, Artyom A. Shestakov, Nikita I. Lapekin, Nikita S. Lazarenko, Pavel B. Kurmashov, Maksim V. Popov

**Affiliations:** 1Department of Chemistry and Chemical Engineering, Faculty of Mechanical Engineering, Novosibirsk State Technical University, 630092 Novosibirsk, Russia; a.shestakov25@mail.ru (A.A.S.); lapekin21@mail.ru (N.I.L.); 9139278003@mail.ru (N.S.L.); kurmaschov@gmail.com (P.B.K.); popovmaxvik@gmail.com (M.V.P.); 2Institute of Chemistry of Solid State and Mechanochemistry, Siberian Branch, Russian Academy of Sciences, 630092 Novosibirsk, Russia; auhina181@gmail.com (A.V.U.); prosanov@mail.ru (I.Y.P.); 3Institute of Inorganic Chemistry, Siberian Branch, Russian Academy of Sciences, 630090 Novosibirsk, Russia; eugene@niic.nsc.ru; 4N.D. Zelinsky Institute of Organic Chemistry, Russian Academy of Sciences, 119991 Moscow, Russia

**Keywords:** expanded graphite, textural characteristics, porosity, surface area, thermal shock, graphite, thermal analysis, heating

## Abstract

Highly porous expanded graphite was synthesized by the programmable heating technique using heating with a constant rate (20 °C/min) from room temperature to 400–700 °C. The samples obtained were analyzed by scanning electron microscopy, energy-dispersive spectroscopy, low-temperature nitrogen adsorption, X-ray photoelectron spectroscopy, Raman spectroscopy, thermogravimetry, and differential scanning calorimetry. A comparison between programmable heating and thermal shock as methods of producing expanded graphite showed efficiency of the first one at a temperature 400 °C, and the surface area reached 699 and 184 m^2^/g, respectively. The proposed technique made it possible to obtain a relatively higher yield of expanded graphite (78–90%) from intercalated graphite. The experiments showed the advantages of programmable heating in terms of its flexibility and the possibility to manage the textural properties, yield, disorder degree, and bulk density of expanded graphite.

## 1. Introduction

Expanded graphite (EG) is used for the production of sealing materials [1], graphite paper [2,3,4], protection coatings [5], fillers for polymer composites [6,7,8], adsorbent [9], electrode material for supercapacitors [10,11], cathodes for aluminum-ion batteries [12], components of antibacterial materials [13,14], adsorbents [15], catalysts [16], etc.

There are two main methods of manufacturing EG, thermal expansion [17,18] and liquid-phase expansion [19]. Thermal expansion is usually used to produce EG from intercalated graphite [20]. Rapid heating leads to the decomposition of intercalated graphite compounds and induces strong formation of a gas phase, which produces a porous cellular structure [21]. The relatively high surface area of the EG determines its applications in adsorption, etc. [22]. In [23], liquid phase expansion was carried out using the mixture of H_2_SO_4_ and H_2_O_2_ (9:1 vol. ratio) and flake graphite.

The porosity of expanded graphite strongly depends on the method of intercalation. For example, van Heerden et al. [24] carried out a comparison of three intercalation methods (FeCl_3_ gas phase intercalation, electrochemical intercalation, and Hummers method) with regard to the characteristics of EG. Gas phase intercalation with FeCl_3_ was carried out using the mixture of graphite and anhydrous FeCl_3_ (1:2 wt. ratio), which were dried (50 °C, 2 h) and heated (300 °C, 25 h). It was found that electrochemical intercalation in sulfuric acid electrolyte and the Hummers method led to the formation of indistinguishable products with similar surface areas of 18.7 and 21.7 m^2^/g, respectively. In [25], the authors carried out heating of graphite intercalation compounds (GICs) from 800 to 900 °C for 20 s, leading to exfoliation of graphite oxide and expanded graphite, but the change in specific surface area was relatively low. In [26,27], it was concluded that the surface area of graphite intercalated with sulfuric acid was low enough (45 m^2^/g). Goudarzi et al. [28] noted that the particle size of GICs and temperature of exfoliation significantly affected the pore size of expanded graphite. Textural characteristics increased when the temperature increased from 700 to 900 °C. The increase in GIC particle size induces a drop in specific pore area of EG.

Usually, the surface area of expanded graphite is relatively low; therefore, novel methods for its synthesis must be developed in order to improve the important characteristics for further application. In this study, programmable heating was used to synthesize EG. The effects of various final temperatures used for heating intercalated graphite on the disorder degree, porosity, and chemical composition of EG were studied in detail. A comparison between thermal shock and programmable heating used for obtaining EG was done.

## 2. Experimental

### 2.1. Preparation of Expanded Graphite

Two sets of highly porous expanded graphite samples were synthesized.

The first set contained 4 samples, which were synthesized from 1 g of intercalated graphite (EG-350-50, Chemical Systems Ltd., Moscow, Russia). To synthesize the thermally expanded graphite and control its porosity, programmable heating, as the first synthesis technique, was used. In addition, a comparison of programmable heating with thermal shock, the second synthesis technique, was done.

The description of samples synthesized under different conditions is shown in Table 1.

Programmable heating was performed by using a certain temperature program set in a muffle furnace (SNOL, Utena, Lithuania) in which the Al_2_O_3_ crucible with intercalated graphite was placed. The sample was heated from room temperature (25 ± 1 °C) to maximum temperatures of 400, 500, 600, and 700 °C in air atmosphere. The heating rate was 20 °C/min, set by the controller of the furnace. The cooled samples were taken from the furnace and were analyzed.

The second method was thermal shock, which was used for comparison with programmable heating. The crucible with the sample was quickly placed into the hot furnace. Usually to synthesize EG, temperatures of 800–1200 °C are used [29,30,31]; therefore, to carry out the comparison, 2 temperatures were used, one lower than this range (400 °C) and one within this range (1000 °C), taking into account that the temperature gradients that influence EG are around 375 and 975 °C/s. Each sample was kept in the furnace for 10 min at the given temperature, then the sample was taken out and cooled down in air atmosphere without additional intensification of cooling.

The appearance of samples obtained by programmable heating and thermal shock are shown in Figure 1.

### 2.2. Characterization

The EG obtained was characterized by the following method. The bulk density of the sample was measured according to GOST 25699.14-93 by compaction of the bed of EG placed into a 10 mL measuring cylinder under the action of vibration until the constant height of the bed was reached.

The yield of EG (%) synthesized using the 2 techniques was determined by weighing the samples using an analytical balance (Sartogosm, Sankt-Peterburg, Russia) as the ratio of sample weight after synthesis to sample weight before synthesis.

The morphology of the surface of samples obtained was investigated by a S-3400N scanning electron microscope (SEM) (Hitachi, Japan). The add-on for energy-dispersive X-ray spectroscopy (EDX) (Oxford Instruments, Abingdon, UK) was used to analyze elements present in the samples.

X-ray diffraction was performed on a DRON-3 diffractometer (Polysorb, Russia) using CuKα radiation (*λ* = 1.54 Å). The number of carbon layers (*n*) in a crystallite along the c-axis direction was calculated by Equation (1) [32,33]:(1)$$n=\frac{Lc}{d_{002}}+1$$
where *L_c_* is the crystallite size and *d*_002_ is the interlayer distance. *L*_c_ parameter was calculated using Sherrer’s formula with values of *K* = 0.9 [34].

The graphitization degree (*g*, %) was calculated by the Maire–Mering Equation (2) [35].
(2)$$g=\frac{3.44-d_{002}}{3.44-3.354}$$

Raman spectroscopy was performed using T64000 Horiba Jobin Yvon (Japan) (*λ* = 514 nm) spectrometer.

The study of the chemical composition of sample surfaces was done using X-ray photoelectron spectroscopy (XPS). The analysis was performed using a surface nano analysis spectrometer (SPECS GmbH, Berlin, Germany) equipped with a PHOIBOS-150 hemispherical analyzer and an XR-50M X-ray radiation source with double Al/Ag anode. Al Kα monochromized radiation (*h** = 1486.74 eV) was used for the study. The relative concentrations of elements in zones of analysis were determined based on the integral intensities of XPS spectral lines, taking into account the section of photoionization of corresponding terms. The spectra were deconvoluted by individual components. After subtracting the baseline by the Shirley method, the experimental curve was decomposed to a set of lines corresponding to the photoemission of electrons from atoms in various surrounding chemicals. CasaXPS software (2.3.24, Teignmouth, U.K.) was used for data treatment. The shape of peaks was approximated with a symmetric function obtained by multiplying the Lorenz and Gauss functions.

The texture characteristics of the obtained carbon materials were studied using low-temperature nitrogen adsorption at 77 K using Quantachrome NOVA 1000 e adsorption unit (Quantachrome Instruments, Boynton Beach, Florida, USA). Specific surface area was calculated using the BET method. The surface area of the pores remaining after filling the micropores with the adsorbate was calculated by the comparative t-method (de Boor and Lippens). Pore size distribution was calculated using the Barrett–Joyner–Halenda (BJH) method. The samples were preliminarily subjected to degassing in a vacuum at 50 Torr for 5 h at 300 °C. The mass of one sample was around 0.01 g.

Thermogravimetry (TG) and differential scanning calorimetry (DSC) were carried out using a STA 449C thermal analyzer (Netzsch, Selb, Germany). The heating was conducted in an Al_2_O_3_ crucible with a tip. The temperature of exfoliation of intercalated graphite was determined in pure argon. The stability of EG samples toward oxidation was analyzed at a heating rate of 10 °C/min in a mixture of argon and oxygen.

## 3. Results and Discussion

Figure 2 shows the TG and DSC curves of intercalated graphite used for preparing EG.

The mass of the sample decreased during heating and there was a sharp drop in the TG curve at 193 °C. According to the DSC curve, the onset of the exothermic peak attributed to exfoliation of the intercalated graphite (as a result of decomposition of graphite intercalation compounds) was 267 °C, and this temperature can be treated as the beginning of thermal exfoliation [36]. The maximum of DSC peak was detected at 360 °C. It was not possible to determine the end of the DSC peak, since the curve was noisy, and this effect was repeated many times with intercalated graphite. It is interesting that the mass loss continued during heating above 400 °C, although the value of loss within a temperature range 400–700 °C was 2.87%. This indicates the occurrence of additional processes in graphite after its exfoliation.

SEM images of the samples subjected to programmable heating and thermal shock are shown in Figure 3 with SEM images of initial intercalated graphite (Figure 3a,b) shown for comparison. Intercalated graphite showed morphology typical for graphite. Heating of intercalated graphite induced the deformation of porous particles (Figure 3c,d) of the sample, with the appearance of cracks on the surface (Figure 3e,f).

Table 2 shows the EDX data of intercalated graphite and TEG1-400. The initial intercalated graphite contained the following elements: mainly C (87.31 at.%), O (11.06 at.%), and S (1.14 at.%), and traces of Al (0.06 at.%), Si (0.18 at.%), Cr (0.05 at.%), Fe (0.07 at.%), and Cu (0.12 at.%).

The presence of sulfur and oxygen is mainly attributed to graphite intercalated compounds, the content of which decreased after thermal expansion of the sample. The composition of intercalated graphite can indicate that the sample is represented by graphite bisulfate. The presence of chromium in the sample makes it possible to suggest that K_2_Cr_2_O_7_ was used as an oxidizer for synthesis. According to data on thermal analysis, the DSC peak attributed to the decomposition of GICs (graphite bisulfate) is exothermic, whereas some studies [37,38] reported that it had an endothermic nature. An exothermic DSC peak appearing as a result of graphite bisulfate decomposition was also found in [39]. Taking into account that the sample expansion was strong enough, there was also small amount of residual elements present in the TEG1-400, attributed to the GICs (S, 0.29 wt.%; O, 5.60 wt.%). The presence of silicon, iron, and copper can be explained by their trace amounts in graphite used for commercial intercalated graphite. The sulfur content in IG is in agreement with EDX data reported in [29].

As can be seen, the EG sample had a porous structure that was more well-developed compared to the intercalated graphite. Figure 4 shows the dependence of the bulk density of EG synthesized by programmable heating on the synthesis temperature. The bulk density of the samples obtained by programmable heating ranged from 0.017 to 0.058 g/cm^3^. Increasing the temperature of synthesis induced a decrease in bulk density; the highest bulk density value (0.058 g/cm^3^) was achieved at 400 °C. In addition, increasing the temperature above 500 °C led to an almost constant bulk density value (0.017–0.018 g/cm^3^) independent of the temperature. Temperatures above 700 °C were not used for the programmable heating method because of strong oxidation of EG beginning from 650–700 °C. The thermal shock method showed similar bulk density values (TEG2-400, 0.016 g/cm^3^; TEG2-1000, 0.019 g/cm^3^), which are lower than that of TEG1-400 (0.058 g/cm^3^) due to higher pressure acting on the structural defects of the material during expansion at relatively high heating rates of ~400 and 1000 °C/s.

The bulk density values were close to the values obtained in [28] for expanded graphite synthesized from GICs with 200 mesh. This mesh size is close enough to the particle size of IG, which was in a range of 100–200 µm.

The data on bulk density, yield, and texture characteristics are shown in Table 3.

One of the important characteristics showing the efficiency of the synthesis process is the yield of EG. Usually, the most of authors do not pay attention to yield, however it is extremely important for chemical engineering and further scale up of technology. The yield was in the range of 77.8–89.2% and decreased with increasing temperature. The highest yield was reached at 400 °C, whereas the bulk density was relatively high, which is inappropriate for some applications (adsorption [40], catalysis [41,42], sealing materials [1], etc.) where this value must be lower. The bulk density of TEG1-500 sample is low enough, but the surface area and yield were lower compared to sample prepared at 400 °C. The difference in yield TEG1-400 and TEG1-500 was 10.2% and it is high enough. Therefore TEG1-400 sample is better compared to other samples from technological and economical points of view.

Specific surface area is linked to bulk density and yield. In particular, the highest surface area of 699 m^2^/g corresponded to TEG1-400, and it decreased with increasing temperature. The bulk density of the samples synthesized at 500–700 °C by programmable heating was almost the same, but the specific surface area changed significantly. The samples obtained using programmable heating and thermal shock were predominantly mesoporous, with a fraction of micropores within 26–41%, and for TEG1-400, TEG1-500, TEG1-600, and TEG1-700, it was 37.3, 26.0, 33.1, and 41.4%, respectively. Thus, increasing temperature induces a decreasing fraction of mesopores. Probably increasing the temperature above 500 °C leads to stronger “opening” of mesopores and micropores, leading to decreased fractions and reduced total surface area. The fraction of both micropores and mesopores in programmable heating decreases with increasing temperature. For thermal shock, the dependence is opposite, i.e., both micropores and mesopores increase.

The differences in porous structure of the samples can be seen in SEM images shown in Figure 3 and Figure 5. The samples obtained using thermal shock look more porous compared to those using programmable heating. However, the data on surface area showed that the difference between the porous structure of samples is significant.

Comparing the textural characteristics of samples obtained by programmable heating and thermal shock, we can see that the samples synthesized by the former had a higher specific surface area than the latter at higher yield. The yield of samples TEG2-400 and TEG2-1000 was 77.4 and 76.9%, respectively, due to the higher driving force of the thermal expansion process at high temperature gradients.

The XRD pattern of the initial commercial intercalated graphite is shown in Figure 6a.

The spectrum is represented by a wide peak around 2θ = 26°, which consists of two main peaks: 2θ = 26.5°, corresponding to graphite (*d*_002_ = 3.368 Å), and 2θ = 26.0°, with an interlayer distance lower than that in pure graphite, corresponding to graphite bisulfate [43]. The EDX and XRD data suggest that the sample of initial intercalated graphite is graphite bisulfate obtained by using K_2_Cr_2_O_7_ as an oxidizing agent. This is also confirmed by XRD data presented in [43], showing a shift of the 002 peak to below 2θ = 26° with an asymmetrical peak appearing in the spectrum. The XRD spectrum of TEG1-400 is represented only by graphite phase and a narrow peak at 2θ = 26.5° (Figure 6b).

There is a link between the number of carbon layers in the crystallite along the *c* direction, the crystallite size *L*_c_, and the specific surface area (Figure 7a). The curves of pore size distribution are presented in Figure 7b–g.

This shows that the specific area decreases with the growth of crystallite size, and the value of *L_c_* = 290 Å corresponds to the highest surface area (699 m^2^/g). The data of average pore size confirm that the pore diameter of EG decreases when increasing temperature using programmable heating. Thermal shock led to the increase in average pore diameter when increasing temperature.

Isotherms of adsorption are presented in Appendix A. Based on the type of adsorption and desorption isotherms, it can be concluded that all EG samples have an inhomogeneous porous structure. All EG samples are characterized by a steep rise of the isotherm in the region P/P_0_ < 0.4 and up to P/P_0_ = 0.99, which indicates the presence of micropores in the structure, which can have a positive effect on the sorption properties of the material. An insignificant loop of the capillary condensation hysteresis of the H4 type is observed when increasing the relative pressure in the region P/P_0_ = 0.4–0.99 due to the presence of slit-shaped mesopores [44]. The isotherms of the analyzed EG samples are difficult to classify according to the generally accepted classification of physical sorption isotherms adopted by IUPAC. The type of analyzed isotherms is most closely matched with IVa type from the proposed updated qualification of isotherms [45]. Type IV is typical for mesoporous adsorbents (e.g., mesoporous molecular sieves). In the case of type IVa isotherms, capillary condensation is accompanied by a slight hysteresis of the H4 type. This occurs when the pore width exceeds a certain critical width, which depends on the adsorption system and temperature, which is typical for cylindrical, narrowed, and slit-like pores with a diameter of ~4 nm.

The FTIR spectra of typical TEG1-400 and IG samples are shown in Figure 8.

The IG and expanded graphite spectra show the presence of O–H vibrations (3500 cm^−1^), C=C stretching vibrations (1642 cm^−1^), and CO stretching vibrations (1750 cm^−1^) [29,46]. The IG spectrum shows an C=S stretching band at 1050 cm^−1^, the intensity of which decreased after exfoliation [47].

Raman spectra and the parameters of spectra of EG are presented in Figure 9 and Table 4. Raman spectrum of IG is presented in Appendix A.

There were D and G bands typical for graphite materials presented in Raman spectra corresponded to disorder in graphite and E_2g_ mode of graphite related to sp^2^ bonded atoms [48]. The position of G peak remained unchanged for the entire set of samples. D and G peaks became narrow when increasing the temperature of graphite expansion. The ratio of intensity of peaks I (D)/I (G) was low enough indicating perfect crystallinity of expanded graphites. The highest I (D)/I (G) ratio attributed to defectiveness of EG was shown by TEG1-400 and it decreased as the temperature rise. The difference of I (D)/I (G) ratio of samples TEG1-400 and TEG2-400 obtained using programmable heating and thermal shock is high enough (0.1 and 0.014, respectively), and it is in agreement with textural characteristics shown in Table 3. The data corresponded to samples obtained by thermal shock showed the deeper removal of GICs from the material making it closer to pure graphite, i.e., material with high graphitization degree. The difference of I (D)/I (G) ratio of TEG1-400 and TEG2-400 (0.1 and 0.014, respectively) samples is in agreement with data of textural characteristics. It was supposed that the higher expansion degree the higher defectiveness. The gas phase formed as a result of decomposition of GICs passes through the certain defect regions in the sample obtained using thermal shock. The number of regions through which the gas phase penetrates in the sample TEG1-400 is higher since its pressure is not so high as in thermal shock. It means that the samples obtained using programmable heating will have higher defectiveness compared to thermal shock because of larger number of defect regions available for gas phase pathways. Higher temperature gradients lead to contribution of only limited defects which were immediately expanded and the subsequent drop in driving force decelerates the expansion.

According to XPS, the typical C1s spectra of IG and EG consist of four main components corresponding to C=C (284.5 eV), C-O (286.5 eV), C=O (288 eV), and O-C=O (288.8 eV) groups (Figure 10) [49,50].

The fractions of these components for IG were 96.7%, 2.8%, 0.3%, and 0.2%, respectively; after exfoliation (TEG1-400), they were 92.9%, 5.6%, 1.1%, and 0.4%, respectively. Exfoliation led to an increase in the C=C component of the C1s spectrum attributed to carbon bonding with carbon in sp^2^ hybridization. It also induced the removal of C-O and C=O groups from the intercalated graphite during heating, whereas the concentration of carboxyls decreased slightly. This confirms the thermodynamic stability of carboxylic groups with respect to heating. However, some articles reported that the temperature range of the release of carboxylic groups was 100–250 °C [51].

The O1s spectrum of intercalated graphite is represented by a peak at 531.5 eV attributed to oxygen in SO_x_ groups, a peak at 532.5 eV attributed to oxygen in C = O groups, and two peaks at 533.6 and 535.1–535.7 eV corresponding to OH groups and water, respectively (Figure 10c). This gives us additional confirmation of the presence of graphite bisulfate in the initial sample used for exfoliation.

The oxidation resistance of expanded graphite is an extremely important characteristic linked to defectiveness, specific surface area, and chemical composition. DSC makes it possible to estimate the dynamics of oxidation of EGs obtained using two methods under consideration. The results of the analysis of DSC data on EG thermal oxidation are shown in Table 5.

It can be seen that the oxidation process of EG has a dependence on the synthesis temperature for the samples obtained using programmable heating. The DSC curves of EG samples are shown in Figure 11.

For example, the TEG1-500 sample showed higher temperature oxidation at the beginning and the highest temperature at the maximum DSC peak. The increased synthesis temperature makes the DSC exothermic oxidation peak broader compared with the relatively narrow peaks for the TEG1-400 and TEG1-500 samples. The TEG1-400 sample showed the lowest heat release during oxidation of 9582 J/g. The EG obtained had a large surface area along with high graphitization degree, accompanied by high onset oxidation temperature, which makes it a good candidate for catalysis (catalyst support for high-temperature processes) and electrochemistry (as a material with good conductivity and porosity).

It is worth noting that the oxidation resistance of the EG obtained was high enough, higher than that of various porous graphites, and can be compared to the expanded graphite coated with SiC (expanded graphite +25 or 100 vol.% silane coupling agent) synthesized in [52], graphite foil obtained from expandable graphite modified with boric acid [53], and boron oxide modified graphite foil reported by Savchenko et al. [54].

According to the data presented above, programmable heating is an alternative way to improve the textural properties and to obtain porous expanded graphite. Table 6 shows a comparison of the specific surface area of expanded graphite obtained in this work with data reported in the literature. As can be seen, the EG synthesized in this study significantly exceeds other samples, and its specific surface area is about 100–200 m^2^/g higher than that of expanded graphite obtained from graphite oxide or graphene oxide, which makes this approach very attractive for practical applications. The high specific surface area of EG obtained by programmable heating can be explained by the relatively low heating rate, which makes it possible for the balloon walls in the cells to expand more gradually, while thermal shock is rapid heating that results in fast formation of expanded cells occurring simultaneously with the formation of defects in the graphite structure. These defects are paths for the gas phase formed as a result of the decomposition of GICs. Since thermal shock has an extremely high heating rate, these defective regions expand rapidly and the gas phase leaves the cell. This effect of “bleeding” of gas leads to the expansion of a certain small number of pores, leaving the remaining pores almost unexpanded. Compared with thermal shock, the heating rate used in programmable heating is low enough that it preserves some defects from rapid expansion and the gas phase stays in the balloon cell. Thus, the number of pores formed is higher and the specific surface area is high enough.

It is worth noting that the heating rate in the programmable heating method is an analogue of residence time, a value usually used in chemical engineering to describe the residence of reagent in a reactor or other apparatus. The higher the heating rate, the lower the residence time, and vice versa.

In conclusion, it can be noted that the method of programmable heating makes it possible to obtain highly porous material at relatively moderate temperatures (400–500 °C) with higher yield. The advantage of this method is the possibility to increase the time of exposure of intercalated graphite in the furnace. At the same time, this method is more flexible than thermal shock because it allows for the possibility to vary the textural properties, yield, and bulk density in a wide range.

## 4. Conclusions

In this study, the highly porous expanded graphite was synthesized by the programmable heating technique using heating with a constant rate (20 °C/min) from room temperature to 400–700 °C. A comparison between programmable heating and thermal shock as methods of producing expanded graphite showed its efficiency at an extremely low temperature (400 °C), and the surface area reached 699 and 184 m^2^/g. It was found that the novel method of production of expanded graphite from intercalated graphite made it possible to obtain a relatively higher yield (78–90%). The EG obtained using programmable heating had a large surface area along with high graphitization degree, accompanied by high onset oxidation temperature, which makes it a good candidate for catalysis and electrochemistry. The programmable heating led to obtaining predominantly mesoporous expanded graphites. The advantage of this method is the possibility to increase the time of exposure of intercalated graphite in the furnace. This method is more flexible than thermal shock because it allows for the possibility to vary the textural properties, yield, and bulk density in a wider range.

## Figures and Tables

**Figure 1 materials-14-07687-f001:**
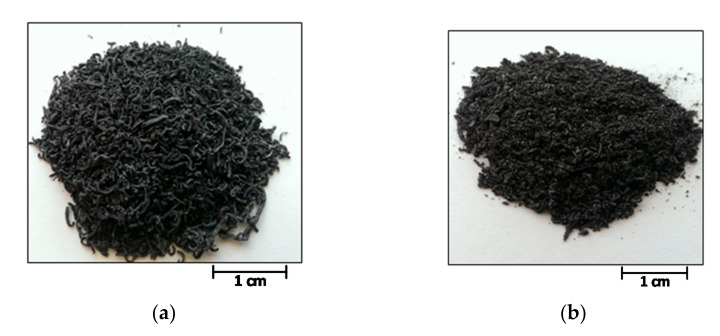
Appearance of the TEG1-400 (**a**) and TEG2-400 (**b**) samples.

**Figure 2 materials-14-07687-f002:**
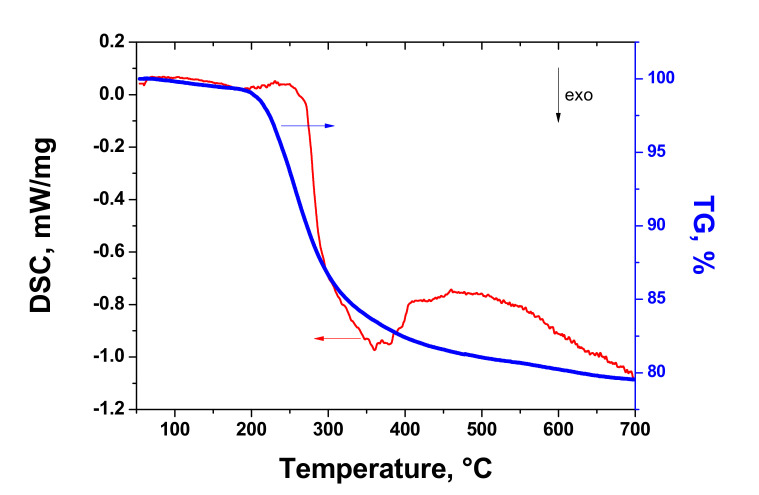
TG/DSC curves of intercalated graphite (Ar, 2.5 °C/min).

**Figure 3 materials-14-07687-f003:**
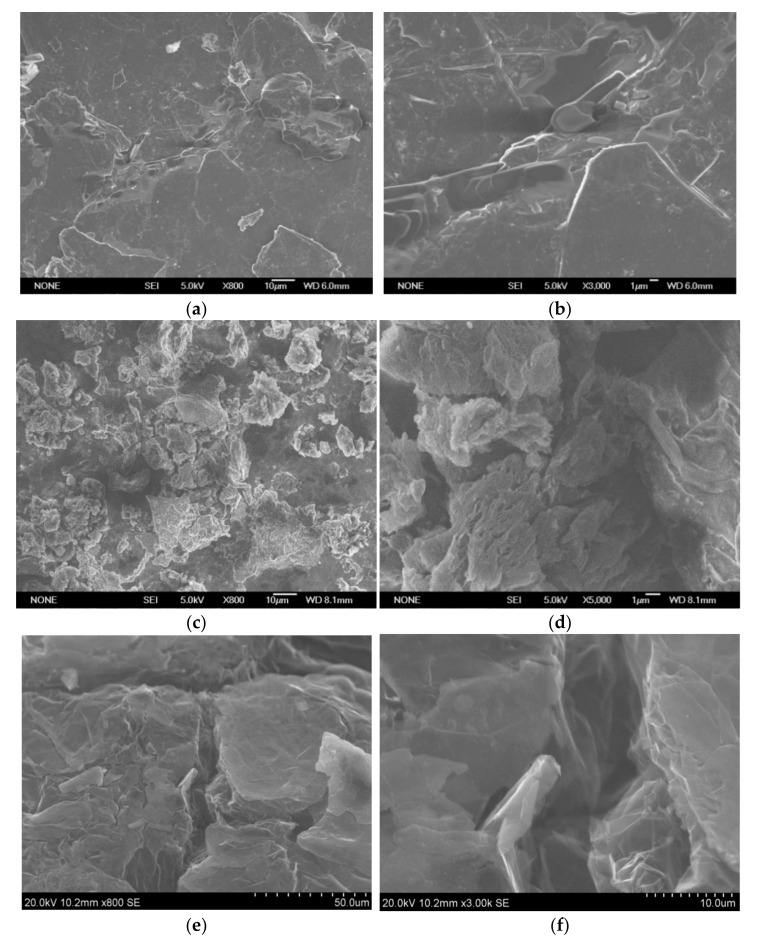
SEM images of the samples: (**a**,**b**)–intercalated graphite; (**c**,**d**)–TEG1-400 sample, obtained by programmable heating; (**e**,**f**)–TEG2-400 sample, obtained by thermal shock.

**Figure 4 materials-14-07687-f004:**
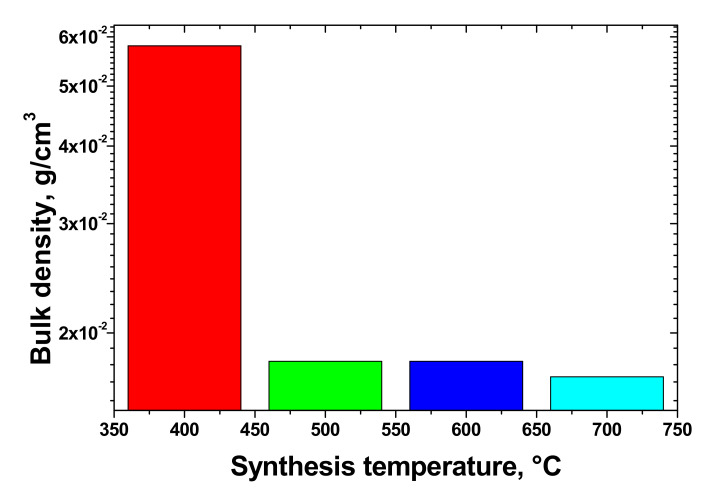
Dependence of bulk density of EG, synthesized by programmable heating, on the synthesis temperature.

**Figure 5 materials-14-07687-f005:**
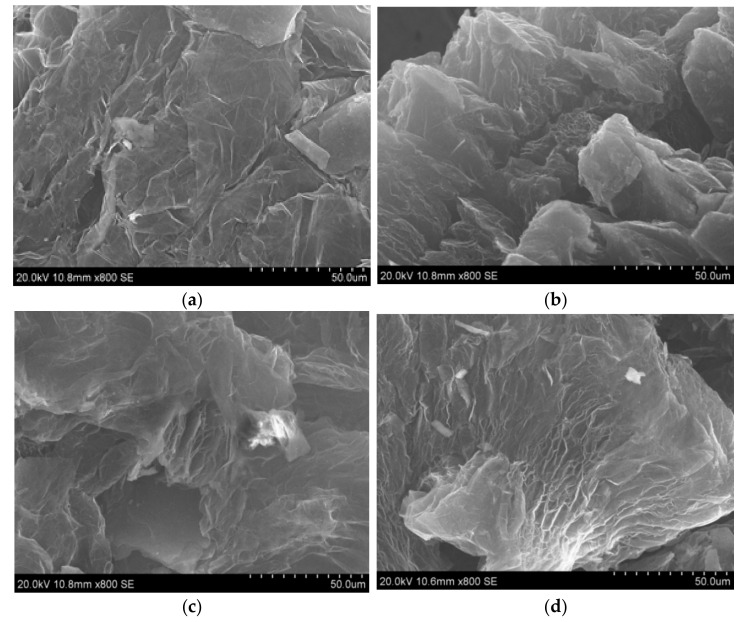
SEM images of the EG samples: TEG1-500 (**a**), TEG1-600 (**b**), TEG1-700 (**c**), TEG2-1000 (**d**).

**Figure 6 materials-14-07687-f006:**
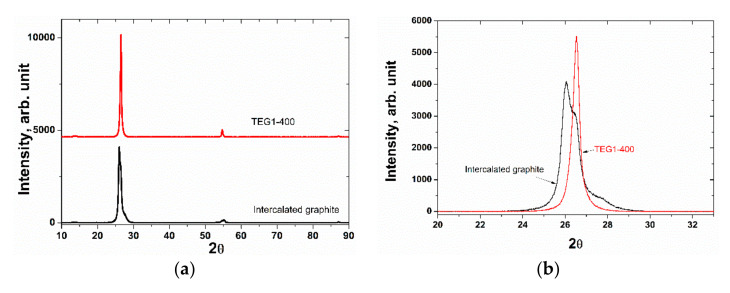
XRD patterns of the intercalated graphite and TEG1-400 (**a**) and 002 reflection of these samples (**b**).

**Figure 7 materials-14-07687-f007:**
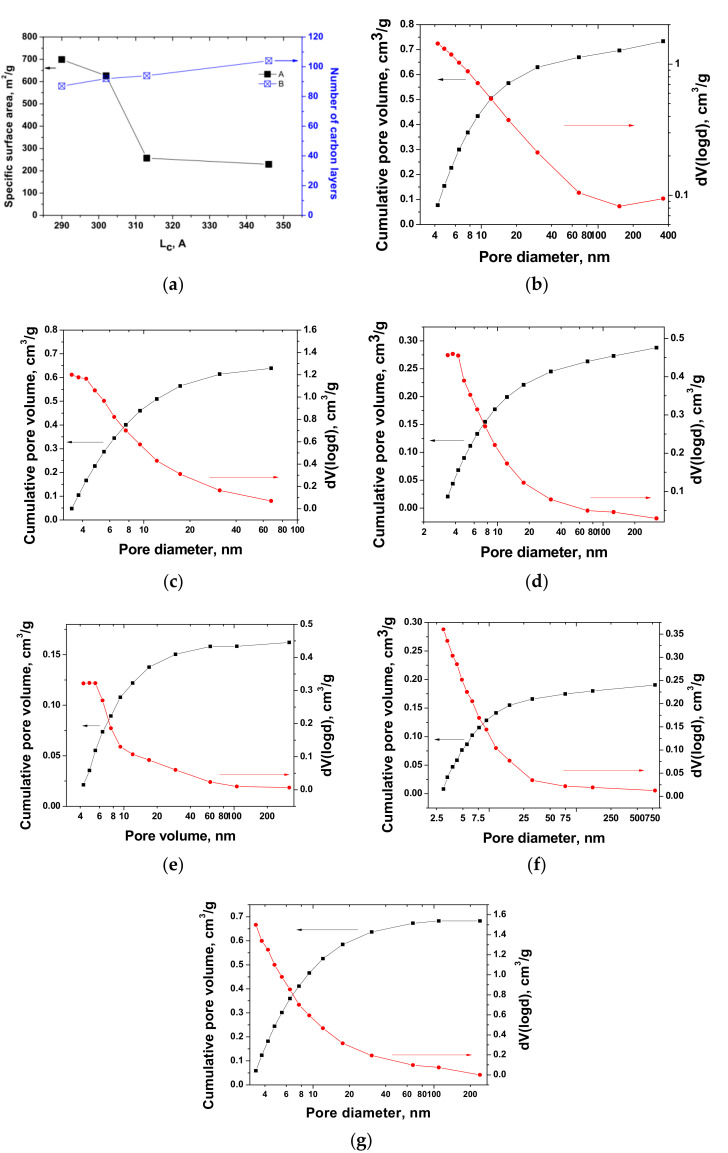
Link between specific surface area, number of carbon layers in crystallite and crystallite width *L_c_*. (**a**), Pore size distribution curves of EG: TEG1-400 (**b**), TEG1-500 (**c**), TEG1-600 (**d**), TEG1-700 (**e**), TEG2-400 (**f**)**,** TEG2-1000 (**g**).

**Figure 8 materials-14-07687-f008:**
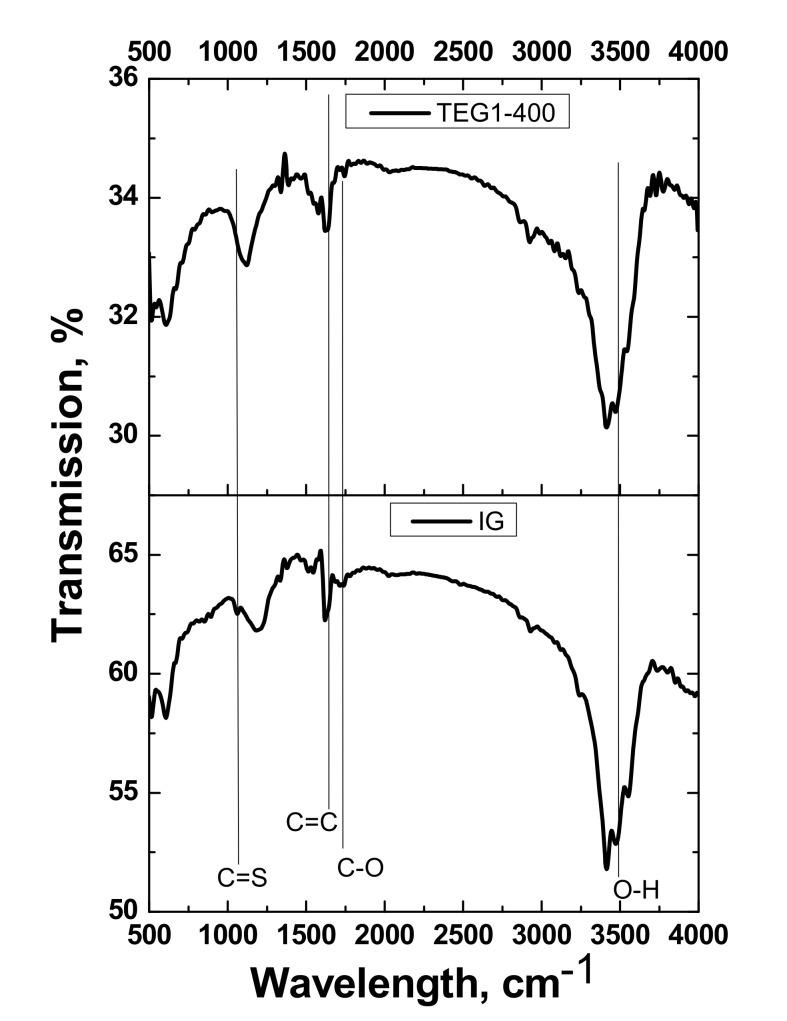
FTIR spectra of IG and TEG1-400.

**Figure 9 materials-14-07687-f009:**
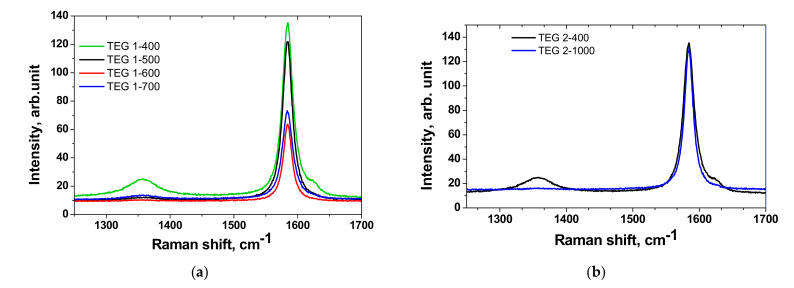
Raman spectra of samples synthesized by programmable heating (**a**) and thermal shock (**b**).

**Figure 10 materials-14-07687-f010:**
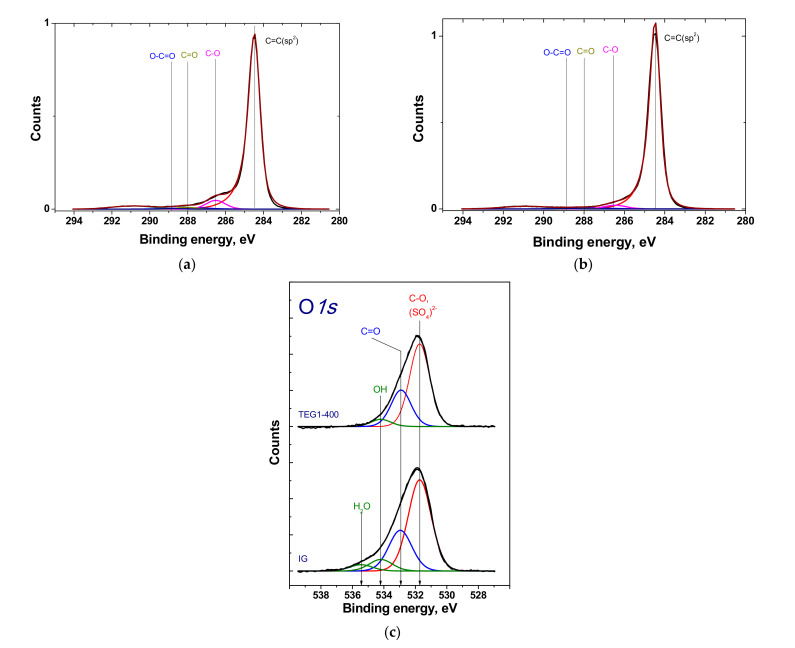
C1s XPS spectra of intercalated graphite (**a**) and TEG1-400 sample (**b**); O1s spectra of intercalated graphite and TEG1-400 sample (**c**).

**Figure 11 materials-14-07687-f011:**
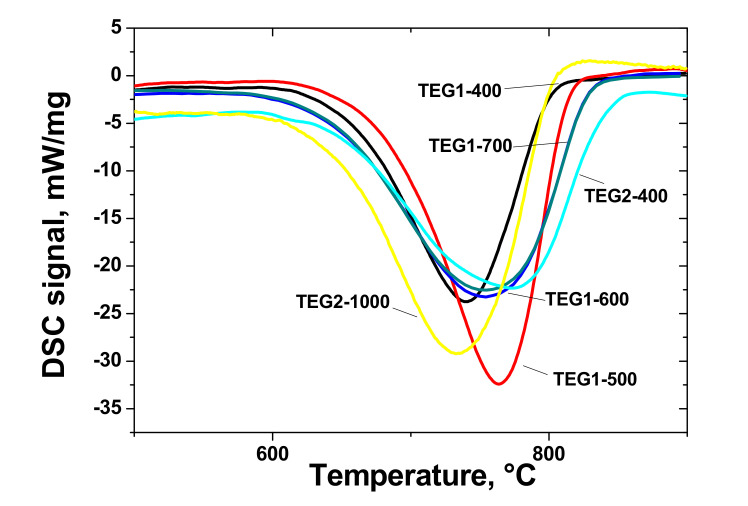
DSC curves of EG samples (10 K/min) (exo down).

**Table 1 materials-14-07687-t001:** Samples synthesized under different conditions.

Sample	Temperature, °C	Synthesis Technique
TEG1-400	400	Programmable heating
TEG1-500	500	Programmable heating
TEG1-600	600	Programmable heating
TEG1-700	700	Programmable heating
TEG2-400	400	Thermal shock
TEG2-1000	1000	Thermal shock

**Table 2 materials-14-07687-t002:** Concentration of the main elements in the EG samples (EDX).

Element	Concentration, at.%						
	Intercalated Graphite	TEG1-400	TEG1-500	TEG1-600	TEG1-700	TEG2-400	TEG2-1000
C	87.31	95.13	96.47	95.69	97.37	93.26	97.46
O	11.06	4.32	3.05	3.75	2.17	6.21	2.14
S	1.14	0.29	0.28	0.23	0.14	0.37	0.05

**Table 3 materials-14-07687-t003:** Yield, bulk density, and textural characteristics of samples synthesized by programmable heating and thermal shock.

Sample	Total Surface Area, m^2^/g	Surface Area of Micropores, m^2^/g	Surface Area of Mesopores, m^2^/g	Total Pore Volume, cm^3^/g	Average Pore Diameter, nm	Yield, %	Bulk Density, g/cm^3^
TEG1-400	699	261	438	0.945	5.4	89.2	0.058
TEG1-500	626	163	463	0.736	4.7	79.0	0.018
TEG1-600	257	85	172	0.325	5.0	78.7	0.018
TEG1-700	229	95	134	0.231	4.0	77.8	0.017
TEG2-400	184	90	94	0.198	4.3	77.4	0.016
TEG2-1000	639	204	435	0.761	4.8	76.9	0.019

**Table 4 materials-14-07687-t004:** Raman spectroscopy data of EG samples.

Sample	D Peak Position, cm^−1^	D_FWHM_, cm^−1^	G Peak Position, cm^−1^	G_FWHM_, cm^−1^	I (D)/I (G)
TEG1-400	1357	53	1584	20	0.1
TEG1-500	1359	19	1584	17	0.0104
TEG1-600	1355	36	1584	16	0.0125
TEG1-700	1357	33	1584	18	0.038
TEG2-400	1357	22	1584	17	0.014
TEG2-1000	1357	22	1584	15	0.007

**Table 5 materials-14-07687-t005:** Analysis of DSC peak of oxidation of EG samples.

Sample	Peak Onset, T_onset_ (°C)	Peak Maximum, T_max_ (°C)	Peak End, T_end_ (°C)	Heat Release, J/g
TEG1-400	652	740	800	9582
TEG1-500	675	764	812	12,207
TEG1-600	639	748	829	12,975
TEG1-700	637	753	828	12,789
TEG2-400	642	773	841	11,967
TEG2-1000	641	733	803	13,773

**Table 6 materials-14-07687-t006:** Surface area values of expanded graphite and related materials reported in literature depending on precursor type, synthesis technique, and synthesis temperature.

Material	Precursor	Specific Surface Area (BET), m^2^/g	Synthesis Technique	Synthesis Temperature, °C	Reference
Expanded graphite	Intercalated graphite	678	Programmable heating (20 °C/min)	400	This work
Expanded graphite	Graphite oxide (modified Hummers method)	400	Thermal shock (1570 °C/min)	500–900	[55]
Expanded graphite	Graphite bisulfate	77	Thermal shock	1300	[1]
Expanded graphite	Graphene oxide(Hummers method)	466	Exfoliation in aqueous media	n/a	[56]
Expanded graphite	Expandable natural graphite flake (interca- lated by using sulfuric acid) from	45	Thermal shock	900	[21]
Expanded graphite	Expandable graphite (H_2_SO_4_–GIC)	10–40	Thermal shock	400–1000	[57]
Expanded graphite	Expandable graphite	22.4	Thermal shock	600	[58]
Sulfur-free Expanded graphite	GIC	245	Thermal shock	950	[9]
Re-expanded EG	Expanded EG	33.5	Thermal shock	800	[59]

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
