# Peer review of "Highly Porous Expanded Graphite: Thermal Shock vs. Programmable Heating"

_materials, 2021, doi:10.3390/ma14247687_

Round 1
Reviewer 1 Report
My comments and suggestions are included in the attachment.

Author Response
Authors would thank the significant work that was done by the reviewer. All the comments were useful and made it possible to extremely improve the quality of the article.
The response is in the file attached.

Reviewer 2 Report
Title: Highly porous expanded graphite: thermal shock vs. programmable heating
Manuscript ID: materials-1470412
Comments:
(1) I would like to see full range of Raman spectrum - intensity of 2D peaks?
(2) How the surface area was measured - description missing.
(3) Gas adsorption measurements? Pore size graph?
Author Response
The comments are in the file attached.

Round 2
Reviewer 1 Report
Title: Highly porous expanded graphite: thermal shock vs. programmable heating
Manuscript ID: materials-1470412
The reviewer values the great efford made by the authors after revision, but I am so sorry and still there are several points that have to be clarified before I can recommend the article to be published in Materials. My comments are listed below:
Abstract:
Page 1, line 21
“678 and 593 m2/g”
Please check de values in accordance with the data of table 3 and correct.
Comments:
Thank you for remark. The data were corrected.
Response:
Abstract:Page 1, line 22
Please, if possible include “, respectively” after “184 m2g-1”
Introduction:
Page 1, line 33.
Could you please introduce a reference related to catalyst applications?
Comments:
The references have been added
Response:
Introduction: Page 1, line 34.
I am so sorry but I cannot see the reference.
Page 2, line 57
“(graphite bisulfate)”. This information should be removed from the introduction section and added to the experimental section.
Comment:
The sentence was removed from Introduction and added to Experimental section.
Response:
Page 2, line 61-62
I would mean just “(graphite bisultate)” should be moved to the experimental section but the rest of the sentence should be stay in the introduction. “The effects of various final temperatures used for heating intercalated graphite on the disorder degree, porosity, and chemical composition of EG were studied in detail.”
Page 7, lines 197-207
The adsorption-desorption N2 isotherms and pore size distribution must be included and discussed.
In addition, the authors should explain, and include in the experimental section, the methods and approximations made for the calculation of total surface area, surface area of micro and mesopores, total pore volume and average pore diameter.
Comments:
The pore size distribution are presented in Figure 7. Isotherms of adsorption are added to Supplementary materials.
Response:
Page 10, Figure 7b
In figure 7b is missing the black and red arrows.
Page10-12 Figures 7b-g
In addition, the curves of pore size distribution are better represented by the pore diameter in logarithmic scale and also with shorter pore diameter (the same for all the figures).
Please check the right axis (dV(logd)) and also include the units.
Page12, lines 278-279:
The figure caption has to be actualized including the pore size distribution curves.
Page 12, lines 285-292
The type of isotherm according to IUPAC classification has to be included.
“M. Thommes, K. Kaneko, A. Neimark, J. Olivier, F. Rodriguez-Reinoso, J. Rouquerol, K. Sing, Physisorption of gases, with special reference to the evaluation of surface area and pore size distribution (IUPAC Technical Report), Pure and Applied Chemistry 87 (2015).”
Additional points:
Arb. Un should be changed by arb. unit in Figure S1.

Author Response
The comments are in the file attached.

Reviewer 2 Report
Authors have improved their paper, it can be accepted to publish now.
Author Response
Authors would thank the reviwer.
